# Exploring Verification Frameworks for Social Choice Alignment

**Jessica Ciupa**                                                        J.E.Ciupa@sms.ed.ac.uk
**Vaishak Belle**                                                               vbelle@ed.ac.uk
*University of Edinburgh*

**Ekaterina Komendantskaya**                              e.komendantskaya@soton.ac.uk
*University of Southampton*

**Editors:** Leilani H. Gilpin, Eleonora Giunchiglia, Pascal Hitzler, and Emile van Krieken

## Abstract

The deployment of autonomous agents that interact with humans in safety-critical situations raises new research problems as we move towards fully autonomous systems in domains such as autonomous vehicles or search and rescue. If autonomous agents are placed in a dilemma, how would they act? The literature in computational ethics has explored the actions and learning methods that emerge in ethical dilemmas. However, our position paper examines how ethical dilemmas are not isolated in a social vacuum. Our central claim in our position paper is that to enable trust among all human users, a neurosymbolic verification of moral preference alignment is required. We propose that the formal robustness properties be applied to social choice modelling. We outline how robustness properties can help validate the formation of stable social preference clusters in deep neural network classifiers. Our initial results highlight the vulnerabilities of models in moral-critical scenarios to perturbations, suggesting a verification-training loop for improved robustness. We position this work as an inquiry into the viability of verifying moral preference alignment, based on our initial results. Ultimately, we aim to contribute to the broader interdisciplinary effort that integrates formal methods, social choice theory, and empirical moral psychology for interpretable computational ethics.

## 1. Introduction

As autonomous systems assume decision-making roles in critical domains, such as search and rescue, autonomous vehicles, and air traffic control, ensuring their adherence to safety constraints remains a growing challenge. While much of formal verification focuses on verifying requirements in safety-critical domains. Our paper proposes a new term, moral-critical, for robustness verification. Moral-critical circumstances arise when an autonomous decision with ethically consequential outcomes. Decisions in these domains may be critical to establishing user trust. Our position paper proposes a neurosymbolic approach, providing formal guarantees for deep learning to model the user's moral norms. Our symbolic component consists of a structured representation of moral dilemmas, with input encoded as labelled choice alternatives from moral preference questionnaires. The input space has defined semantic constraints constructed by standard error (SE). This symbolic element defines the geometric bounds against which we verify our deep neural network (DNN) classifiers. The application of robustness logic to cultural embeddings represents a novel approach that enables formal verification of geometric boundaries in high-dimensional spaces.

While clustering and threshold techniques in behavioural simulations are effective for descriptive grouping, they lack the capacity to verify robustness against perturbations. Having formal verification in embedded spaces of social behaviours in groups could provide provable guarantees that the embeddings within the bounds are classified as belonging to group norms. Moral preferences in this paper are the individual-level choice data, and the emergent social norms are from aggregated culture level structures. We aim to critically examine the feasibility and necessity of neurosymbolic verification for the deployment of ethical AI. We advocate for a research agenda to bridge social choice theory, computational modelling, and formal methods, outlining key challenges and focusing on the open research questions in bridging formal methods with moral social choice reasoning.

## 2. Related Works

The concept of social preferences in group contexts in computational ethics (Kleiman-Weiner et al., 2017; Kim et al., 2018) is based on the idea that there is no universal code of ethics and that user preference in moral choice is key to deployment in domains with an impact on an individual's personhood and morality, which we have termed moral-critical. Ensuring ethical compliance in autonomous systems requires interpretable and verifiable models, particularly in high-stakes domains. While deep learning models have demonstrated impressive predictive capabilities, their lack of interpretability and formal guarantees poses a challenge for deployment in moral dilemmas. Traditional robustness verification methods, including the use of geometric bounds, constrain acceptable input-output behaviours to predefined regions of the feature space (Casadio et al., 2022; Flood et al., 2025). For example, Casadio et al. (2022) demonstrates how formal properties, such as invariance or bounded perturbation response, can be embedded into training objectives, creating a feedback loop between training and verification with tools like Marabou (Wu et al., 2024) that verify robustness at scale.

For further development in the verification of moral-critical domains, which applies when moral reasoning spans both discrete symbolic logic and continuous inputs, the market of hybrid system verifiers, such as KeYMaeraX, CORA, and Vehicle, suggests exciting exploration (Daggitt et al., 2023; Teuber et al., 2024; Althoff, 2015). These tools enable reasoning about systems that operate at the intersection of logical planning and continuous control, which is characteristic of morally autonomous agents. Despite their theoretical suitability, the literature on applying these models to social choice is limited. This paper proposes the building blocks for their application to moral reasoning and potential cases involving evolving or context-dependent values, which can be meaningfully explored.

## 3. Verification of Classified Group Norms

We extend the robustness verification framework (Casadio et al., 2022; Flood et al., 2025) to the domain of emergent social preferences. While robustness in neural networks is defined in terms of stability under small, norm-bounded perturbations, such as $\epsilon$-ball constraints around an input, this framework is extended in order to capture morally relevant variation observed in human cultural judgments. Therefore, to evaluate semantic stability across culturally grounded input variability we compare two types of verification bounds:

1. **Epsilon-ball robustness**, representing traditional, tightly bounded perturbations useful for adversarial threat models.

2. **Semantically grounded geometric robustness**, where verification regions are defined by *hyperrectangles* constructed from the *standard error (SE)* of culturally clustered groups in the Moral Machine dataset (Awad et al., 2018).

These hyperrectangles encode bounded moral diversity observed across cultural clusters. For example, a Western cluster with a high SE produces a larger region, capturing the broader internal disagreement among its members. The output provides a form of cultural robustness that better aligns with ethical generalisation requirements of regions. We aim to offer a more interpretable and socially relevant extension to robustness verification: one that allows for bounded behavioural guarantees under meaningful, morally salient perturbations, rather than abstract geometric constraints alone.

**Definition 1 (Classification Robustness (CR))** *Let $N : \mathbb{R}^n \to \mathbb{R}^m$ be a neural network classifier. Let $(\hat{x}, y)$ be a data point, where $y$ is the true label associated with $\hat{x}$, and let $\epsilon > 0$ be a perturbation bound. Then $N$ is classification-robust around $\hat{x}$ if:*

$$CR(\epsilon, \hat{x}, y) \equiv \forall x : \|x - \hat{x}\| \leq \epsilon \Rightarrow \arg\max N(x) = y$$

*In other words, for all inputs close to $\hat{x}$, the classifier's predicted label remains equal to the label of $\hat{x}$.*

Following the principles laid out in Casadio et al. (2022), we will utilise the same CR verification workflow. This reframes group norm classification as a verifiable, constraint-driven task rather than a purely empirical statistical one.

## 3.1. Additional Robustness Properties

While CR is discrete and interpretable, it is non-differentiable and may fail near decision boundaries. We therefore define additional, complementary robustness notions:

- **Standard Robustness (SR):** *Requires small input changes to induce small output changes:*

$$SR(\epsilon, \delta, \hat{x}) \equiv \forall x : \|x - \hat{x}\| \leq \epsilon \Rightarrow \|f(x) - f(\hat{x})\| \leq \delta$$

  *where $f(x) \in \mathbb{R}^m$ is the vector of output scores over all classes, and $\delta > 0$ bounds the change.*

- **Lipschitz Robustness (LR):** *A stronger condition that linearly bounds output variation by input variation:*

$$LR(\epsilon, L, \hat{x}) \equiv \forall x : \|x - \hat{x}\| \leq \epsilon \Rightarrow \|f(x) - f(\hat{x})\| \leq L \cdot \|x - \hat{x}\|$$

  *where $L$ is the Lipschitz constant controlling the sensitivity of $f$.*

- **Strong Classification Robustness (SCR):** *A differentiable approximation to CR that enforces a minimum confidence in the correct class:*

$$SCR(\epsilon, \eta, \hat{x}, y) \equiv \forall x : \|x - \hat{x}\| \leq \epsilon \Rightarrow f_y(x) \geq \eta$$

  *where $f_y(x)$ is the predicted score (e.g., softmax probability) assigned to the true class $y$, and $\eta \in [0, 1]$ is a confidence threshold.*

### 3.2. Robustness Metrics

Each robustness property trades off three key dimensions: **interpretability**, **logical strength**, and **verifiability**. Table 1 summarizes the relative strengths of four conditions: Constant Robustness (CR), Soft Constant Robustness (SCR), Soft Robustness (SR), and Lipschitz Robustness (LR).

Table 1: Comparison of interpretability, logical strength, and verifiability

| Condition | Interpretability | Logical Strength | Verifiability |
|---|---|---|---|
| CR | **Highest** (label constancy) | Weakest | Low (non-differentiable) |
| SCR | High (soft version of CR) | Moderate | High (differentiable) |
| SR | Moderate (output similarity) | Strong | High (differentiable) |
| LR | Lower (based on distances) | **Strongest** | High (Lipschitz-encodable) |

Interpretability reflects how intuitively meaningful the condition is to a human reader. Logical strength refers to how strongly one condition implies others. Verifiability indicates whether the property can be encoded into differentiable optimization or SMT solving.

### 3.3. Preliminary Results

While this position paper is primarily conceptual, we conducted preliminary experiments to demonstrate the feasibility of applying formal verification techniques to moral decision-making systems. Our aim is to encourage broader exploration of verifiability in domains not traditionally subject to formal guarantees, such as moral AI.

We trained a simple feed-forward neural network to classify countries into three moral-cultural clusters: **Western**, **Southern**, and **Eastern**. Clusters were derived using Agglomerative Clustering on AMCE features provided in `CountriesChangePr.csv` (Awad et al., 2018). Experiments revealed that even a simple, high-performing neural network failed most verification tests, with only one property was verified at $\epsilon = 0.01$ in Table 2.

Table 2: Verification results using Marabou across clusters and perturbation strategies

| Cluster | Perturbation Type | $\epsilon$ | Result |
|---|---|---|---|
| Western | Uniform Box | 0.01 | Verified (2/2 safe) |
| Southern | Uniform Box | 0.01 | Counterexample Found |
| Eastern | Uniform Box | 0.01 | Counterexample Found |
| Western | Uniform Box | 0.1 | Counterexample Found |
| Southern | Uniform Box | 0.1 | Counterexample Found |
| Eastern | Uniform Box | 0.1 | Counterexample Found |
| Western | Semantic (Hyperrectangle) | SE-scaled | Counterexample Found |
| Southern | Semantic (Hyperrectangle) | SE-scaled | Counterexample Found |
| Eastern | Semantic (Hyperrectangle) | SE-scaled | Counterexample Found |

## 4. Discussion

Formal verification emerges as a promising yet underutilised tool in assessing the reliability of social decision-making processes in autonomous systems. The symbolic layer of analysis enables stronger claims about why certain autonomous decisions align or fail to align with culturally defined moral preferences. The preliminary results exhibited the lack of robustness across cultures and validated the core hypothesis that culturally variant data clusters require attention. Specifically, clusters with lower representation in the data. We first verified the classifier for $\epsilon = 0.01$, assuming this small perturbation would yield positive verification results. Robustness was confirmed only for the Western cluster, likely due to its greater representation in the training data. In contrast, Marabou found counterexamples for both the Eastern and Southern clusters, revealing vulnerabilities in the model. The larger bound $\epsilon = 0.01$ and nuanced Hyperrectangle verification approach also found vulnerabilities in the model.

These counterexamples could involve a minimal perturbation in the input that flipped the model's decision. An example from the Southern cluster is a scenario where an autonomous vehicle must choose between saving a child pedestrian or an elderly passenger. The classifier, trained on culturally mixed data, receives a minor perturbation to just one feature, such as slightly reducing their preference for saving younger individuals, causes the classifier to flip its prediction from Southern to Western. Although the change in input is minimal, its moral significance is considerable: the decision reverses in a way that directly violates the group norm. Group preferences are not just statistical patterns; they represent a form of collective moral will, aggregating individual preferences to protect decision-makers from guilt and social responsibility. If the model fails to uphold this aggregated choice under realistic cultural variation, it introduces moral instability into the system. In this case, the model's vulnerability is not merely a technical failure but an ethical one; it undermines the legitimacy of the decision from the perspective of the society in which it is deployed.

Our results suggest that future work should integrate verification-guided training strategies to enforce region-specific and value-sensitive robustness. Models must not only be robust in an adversarial sense but must also remain stable under socially plausible shifts in moral input. One promising direction is to combine robustness verification with model interpretability techniques such as LIME or SHAP. These tools can help determine whether the model is learning meaningful and human-aligned representations or merely overfitting to statistical regularities in overrepresented clusters.

## 5. Conclusion

The position paper proposed viewing group norms as properties to be verified, quantified, and tested for the stability of social group boundaries in embedding spaces. While still in its early stages, the synergy between formal verification and computational social choice is poised to make a significant contribution to future AI ethics research, ensuring that as social norms shift, technologies remain not only verified but also interpretable in terms of moral alignment. This opens utilising features in empirical moral psychology in neurosymbolic approaches for computational ethics, highlighting the interdisciplinary efforts required for morally aligned autonomous agents.

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

## 6. Supplementary Material

### 6.1. Verification Output

For $\epsilon = 0.01$:

```
Verifying properties:
  property−western [...............................] 2/2 queries
    result: − Marabou proved no counterexample exists
Verifying properties:
  property−southern [...............................] 0/2 queries
    result: X − Marabou found a counterexample
      x: [ 1.0e−2, 1.0e−2, −4.654e−3, 1.0e−2, −1.0e−2,
           −1.0e−2, 1.104e−3, 9.4e−5, −1.0e−2 ]
Verifying properties:
  property−eastern [...............................] 0/2 queries
    result: X − Marabou found a counterexample
      x: [ 1.0e−2, 1.0e−2, −1.0e−2, 1.0e−2, −1.0e−2,
           −1.0e−2, −1.0e−2, 5.695e−3, −1.0e−2 ]
```

For $\epsilon = 0.1$:

```
Verifying properties:
  property−western [...............................] 1/2 queries
result: X − Marabou found a counterexample
      x: [ 5.3715e−2, −1.5213e−2, 4.1891e−2, 6.869e−3, −2.8956e−2,
           −1.7541e−2, −2.8409e−2, 4.5439e−2, −4.2906e−2 ]
Verifying properties:
  property−southern [...............................] 0/2 queries
    result: X − Marabou found a counterexample
      x: [ 2.3003e−2, 2.3255e−2, −1.36e−2, −9.61e−4, 9.787e−3,
           −8.395e−3, 2.0861e−2, 9.66e−3, −2.4799e−2 ]
Verifying properties:
  property−eastern [...............................] 0/2 queries
    result: X − Marabou found a counterexample
      x: [ 2.3003e−2, 2.3255e−2, −1.36e−2, −9.61e−4, 9.787e−3,
           −8.395e−3, 2.0861e−2, 9.66e−3, −2.4799e−2 ]
```

Hyperrectangle from SE:

```
Verifying properties:
  property−western [...............................] 0/2 queries
    result: X − Marabou found a counterexample
```

```
     x: [ 4.8541e−2, −1.3644e−2, 3.7494e−2, 2.2753e−2, −1.0351e−2,
          −1.2661e−2, −2.2838e−2, 2.0656e−2, −5.0068e−2 ]
Verifying properties:
  property−southern [...............................] 0/2 queries
     result: X − Marabou found a counterexample
       x: [ 3.1378e−2, 2.2682e−2, 3.5604e−2, 4.151e−3, −7.506e−3,
            −8.681e−3, 1.92e−2, 1.5135e−2, −6.4713e−2 ]
Verifying properties:
  property−eastern [...............................] 0/2 queries
     result: X − Marabou found a counterexample
       x: [ 3.449e−2, 1.3795e−2, −1.6207e−2, −3.9535e−2, 1.085e−2,
            −9.544e−3, 2.4348e−2, 2.2493e−2, 2.273e−2 ]
```

