# OpenReview forum: "Exploring Verification Frameworks for Social Choice Alignment"
_nesyconf.org/NeSy/2025/Conference_Phase_2 — NeSy 2025 - Phase 2 Poster_

### Official Review · Reviewer_Pfu8 · 2025-07-05
**A more concrete but still not very convincing proposal for verifying social alignment**

**Rating:** 5
**Confidence:** 4

**Review:**

The paper has been significantly revised into a more concrete position piece with welcome preliminary results. However, its core proposal - that verifying the geometric robustness of a classifier is a meaningful proxy for alignment with 'social norms' - remains a significant and not a very convincing conceptual leap.

Pros:
- The paper is well-written and easy to follow.
- The addition of a concrete experimental setup using the Moral Machine dataset is a great improvement.
- It frames an intriguing and important research challenge.

Cons:
- The link between the proposed method (verifying robustness in an epsilon-ball) and the high-level goal (aligning with social norms) is still tenuous at best.
- The approach treats the model as a black box, so it cannot guarantee any meaningful or interpretable representation of norms has been learned.
- Claims about achieving "interpretable" moral alignment are overstated, as the method only checks behavioral robustness, not interpretability.
- The paper's own results, which show verification mostly failing, underscore the task's extreme difficulty more than they validate the proposed path forward.
- The evaluation setup is very simplistic, significantly lagging behind the greatness of the claimed concept.

Overall:
While this is a clearly improved contribution, its core premise remains unconvincing to me. The paper successfully proposes a research direction but falls short of making a compelling case that its specific method is a viable path toward truly aligning AI with complex human values. It provides some valuable references and ideas to think about, though.

**Anonymity:**

Remain anonymous

---

### Official Review · Reviewer_GG2G · 2025-07-07
**Timely problem, but unclear contribution**

**Rating:** 4
**Confidence:** 4

**Review:**

**Summary**

This is a position paper that establishes a research agenda for using Machine Learning formal verification tools to verify that deep learning systems satisfy certain moral preferences. The paper introduces some preliminary methodology to achieve this and reports briefly on some preliminary experiments.

**Strengths**
- _Relevance_ The paper addresses the topic of verifying that an AI system behaves in accordance to certain moral preferences.
- _Clarity_ The paper is written in good English and is easy to follow.

**Weaknesses**
- _Relevance to NeSy_ It is not clear what aspects of the proposed approach are symbolic.
- _Contribution_ It is not very clear what is the contribution of the paper. As far as I understand, the paper describes existing measures of robustness of an ML system and suggests that these could be applied to situations where moral preferences are involved. It also reports briefly on preliminary experiments. If the main contribution is an empirical evaluation, then Section 3.3 should be much larger. If the main contribution is the idea that robustness checks can be used in a moral setting, I would expect to see a more clear description of the real-world task at hand (what do the inputs represent, what do the outputs represent, how does all this precisely relate to social choice?), and some discussion about why this is a novel and promising idea. Alternatively, I would expect a detailed discussion for why these specific robustness checking techniques are suited to the particularities of this specific problem.
- _Technical Quality_ The formulas presented are unclear (see detailed comments below); the datasets used in the preliminary evaluation are also unclear, and I cannot see what is the relation between the ML task related to social choice and the classification robustness models considered in this paper.

**Resolution**

I am afraid that after reviewing this paper a second time, I still have trouble understanding what is the contribution of the paper. I believe the topic of verifying ML systems working on domains related to ethical preferences is very timely and very suited to this conference. But it is hard to see *what* exactly is being proposed in this paper beyond the general claim that using robustness analysis could help. I would really like to see a version of this paper where the ML tasks related to social choice are clearly explained, and where the paper describes clearly the reason for using classification robustness techniques, how they help, and what are the specific challenges in this setting.

**Further comments**

The abstract is not very precise; this makes it hard to follow what are the motivations and the contribution.

Section 1. The definition of "moral-critical" is unclear: what does it mean to "impacts an individual’s ethical and moral perspective"? Why should the system take the moral preferences of the human that it interacts with into account? (I'm not suggesting that there aren't any reasons, but I think that the existing reasons must be written explicitly here)

Section 1. The introduction switches between talking about moral preferences and social norms. It should clarify how these are related. Similarly, it jumps from talking about verification, to talking about robustness.

Section 1. "Clustering and threshold techniques in behavioural simulations are effective for descriptive grouping but not for verifying geometric boundaries" This will be hard to understand for the audience of NeSy. Could you make this more self-contained, and/or add references?

Page 3: please provide the meaning of DNNs after the first use.

Page 3: "in our setting, such perturbations can be interpreted as social cues that deviate from dominant cultural norms" - I felt this was too abstract for me to evaluate the importance of such perturbations. Could you provide an example, please?

Page 3: I do not understand the definition of CR(\epsilon, \hat{x}): arg max N(x) is presumably the x that maximises N(x), but here x is quantified already at the start of the formula. Furthermore, why are we considering a max, and not a min, or being exactly y? Similarly, I cannot understand the last Robustness Property of section 3.1: why is f_y(x) greater than \eta? What is "the output score for class y?"

Page 3: what are the inequalities CR > SCR > ... ?


_Minor Issues_


P.3 "a continuous verification cycles" -> cycle

**Anonymity:**

Disclose identity

---

### Official Review · Reviewer_WV87 · 2025-07-08
**The paper is suitable for publication with minor revisions to address and enhance the discussion of results and interdisciplinary connections.**

**Rating:** 7
**Confidence:** 5

**Review:**

The authors have successfully addressed the concerns raised in Review 1, Review 2, and the Meta Review. They narrowed the scope to focus on verifying moral preference classification using robustness techniques (ε-balls and hyperrectangles), provided a clear verification pipeline, and presented preliminary results to support their claims. The paper maintains an interdisciplinary approach, replaces loose terms with rigorous methods, and ensures readability, avoiding bold claims or a “blog post” style.

1. The paper could strengthen its contribution by emphasizing the central claim in the abstract/conclusion, expanding the discussion of preliminary results, and providing deeper interdisciplinary connections.
2. The term “neural-symbolic” should be corrected to “neurosymbolic.”
3. The supplementary material (Section 6) could be better integrated into the main text to avoid key details.
4. The results could be expanded slightly to strengthen the empirical contribution. For example, by discussing the implications of the Western cluster’s robustness versus the failures of the Southern and Eastern clusters.

**Anonymity:**

Disclose identity